# Incidences of Laryngospasm Using a Laryngeal Mask Airway or Endotracheal Tube in Paediatric Adenotonsillectomy: A Systematic Review

**DOI:** 10.3390/jcm14103369

**Published:** 2025-05-12

**Authors:** Kevin Zi Kai Ooi, Rufinah Teo, Kok-Yong Chin

**Affiliations:** 1Department of Anaesthesiology and Intensive Care, Faculty of Medicine, Universiti Kebangsaan Malaysia, Cheras 56000, Malaysia; kevinooizikai@yahoo.com; 2Department of Pharmacology, Faculty of Medicine, Universiti Kebangsaan Malaysia, Cheras 56000, Malaysia; chinky@ukm.edu.my

**Keywords:** laryngospasm, paediatric, adenotonsillectomies, laryngeal mask airway, endotracheal intubation

## Abstract

**Background/Objectives:** Adenotonsillectomy is common in paediatric otorhinolaryngology. Endotracheal intubation (ETT) has long been the preferred technique for securing the airway during anaesthesia, while the laryngeal mask airway (LMA) was introduced later as an alternative option. However, it is still unclear which of these airway management methods is associated with a lower risk of triggering laryngospasm. This systematic review compares incidences of laryngospasm between the LMA and ETT in paediatric adenotonsillectomy. **Methods**: This systematic review followed guidelines outlined by Preferred Reporting Items for Systematic Reviews and Meta-Analyses (PRISMA). An extensive search of the literature was performed across three electronic databases, i.e. PubMed, Scopus, and Web of Science, covering all records available up to February 2024. Original studies comparing the use of LMAs and ETT in adenotonsillectomy among paediatric patients (aged 1 month–18 years) and reporting incidences of laryngospasm as the primary outcome were included in this review. **Results**: Five studies were included in the current review: three randomised controlled trials (RCTs) and two retrospective cohort studies. Incidences of laryngospasm during the use of LMAs and ETT in paediatric adenotonsillectomy were equivalent in most of these studies. **Conclusions**: The LMA does not reduce the incidence of laryngospasm as compared to ETT in paediatric adenotonsillectomy. More RCTs should be conducted to validate this observation.

## 1. Introduction

Adenotonsillectomy ranks among the most commonly performed surgeries in paediatric otorhinolaryngology [1,2]. It is typically indicated for conditions such as nasal breathing difficulties, recurrent otitis media, repeated upper airway infections, and obstructive sleep apnoea [3,4]. Managing the airway in children undergoing this procedure poses distinct challenges. Although endotracheal intubation (ETT) has been the conventional method for paediatric anaesthesia for many years, the laryngeal mask airway (LMA) has gained popularity as an alternative airway device since its introduction in 1983 [5].

Invented in 1983, the LMA was introduced into clinical practice and has been an alternative tool for airway management. The LMA forms a low-pressure seal above the laryngeal inlet. Since its introduction, the LMA has been used in multiple short surgical procedures, avoiding the need to paralyse and intubate the trachea [6]. Numerous studies showed equal, if not fewer, complications with the use of LMAs [7,8,9,10,11,12,13]. The airway involved in the surgical field is more prone to laryngospasm in children [14,15]. However, there have been conflicting results on whether the usage of the LMA or ETT is superior to the other in terms of incidences of laryngospasm during adenotonsillectomy [8,11,13,16,17]. Laryngospasm is a sudden and forceful closure of the vocal cords that can result in severe respiratory distress and rapid desaturation if not promptly managed. The incidence of laryngospasm in children ranges from 4% to 14% [16].

The laryngeal mask airway (LMA) offers several general benefits, such as easier insertion that does not require direct visualisation of the larynx [5], diminished respiratory and cardiovascular reflex responses [10], a lower chance of causing injury within the larynx, and a shorter duration needed for extubation, and avoids the need for muscle relaxants and reversal agents and the potential side-effects of these drugs [8]. For many short low-risk intraoperative procedures, the LMA has been shown to be superior to ETT in terms of anaesthesia-associated complications, such as laryngospasm and an increase in the surgical turnover rate [10]. Nonetheless, Verghese and Brimacombe reported that when spontaneous ventilation was maintained, the use of LMAs did not lead to a significant decrease in anaesthesia-related critical events compared to ETT. These events included regurgitation, vomiting, aspiration of stomach contents, bronchospasm, laryngospasm, gastric distension, low blood pressure, arrhythmias, and cardiac arrest [10].

The use of ETT in adenotonsillectomy provides adequate surgical access and reduces overall incidences of laryngospasm. However, ETT requires additional usage of neuromuscular blocking agents during placement, which could contribute to postoperative respiratory complications such as atelectasis and prolonged operation room time, impacting the case turnover rate due to the time required for extubation [8]. Furthermore, the use of laryngoscopy during intubation could trigger cardiovascular and respiratory reflexes, including laryngospasm.

Since the introduction of anaesthesia in 1846, control and airway protection have been crucial considerations while a patient is under anaesthesia. During the initial stages, ETT has been the standard for adenotonsillectomy [18,19,20,21,22]. However, complications of ETT include trauma, cardiovascular stimulation, endobronchial or oesophagal intubation, coughing, bronchospasm, laryngospasm, and laryngeal oedema with airway obstruction after extubation. A Boyle–Davis (B–D) gag helps open the mouth to ensure the surgical site is adequately exposed. Furthermore, the possibility of the ETT tube being kinked and compressed by the B-D gag tongue blade, as well as premature extubation during manipulation of the B-D gag, may occur, which could be disastrous [23,24,25]. Both methods have their advantages and limitations, leading to ongoing debates among anaesthesiologists regarding their optimal usage in this specific context.

This study aimed to systematically investigate and compare the use of the LMA and ETT in paediatric adenotonsillectomies, particularly regarding laryngospasm. This objective was achieved by examining the relevant literature and analysing clinical data provided. The findings of this study shall contribute to the further enhancement of the anaesthetic safety profile of paediatric adenotonsillectomies.

## 2. Materials and Methods

This systematic review was conducted according to Preferred Reporting Items for Systematic Reviews and Meta-Analyses (PRISMA) (Appendix A). The protocol of this study was registered in the International Platform of Registered Systematic Review and Meta-analysis Protocols, registration number INPLASY202460083 (DOI: 10.37766/inplasy2024.6.0083).

A comprehensive literature search was conducted using three electronic databases, including PubMed, Scopus, and Web of Science, from their inception until February 2024. The following search string was used: (“Laryngospasm” OR “Laryngeal Spasm”) AND (“Paediatrics” OR “Pediatrics” OR “Children”) AND (“Adenotonsillectomy” OR “Adenotonsillectomies”) AND (“Laryngeal mask airway” OR “Laryngeal mask” OR “Supraglottic Airway”) AND (“Endotracheal tube” OR “Intubation”). A manual search was performed to retrieve additional records from the reference list of included studies and relevant review papers.

Two independent reviewers (K.Z.K.O. and K.-Y.C.) screened the titles and abstracts of the literature found, followed by full-text screening. They identified studies based on predefined inclusion and exclusion criteria. Any discrepancy was resolved by discussion with the third author (R.T.).

Studies with the following characteristics were included: (1) randomised controlled trials (RCTs) and cohort studies; (2) involved paediatric patients (aged 1 month–18 years) who underwent adenotonsillectomy procedures under general anaesthesia; (3) compared the use of the LMA and ETT in adenotonsillectomy; (4) reported incidences of laryngospasm as the primary outcome; and (5) published or translated into the English language. Studies excluded had the following characteristics: (1) lacked sufficient data on the outcomes relevant to the research question; (2) did not compare the LMA and ETT directly; and (3) did not separate the paediatric population from the overall cohort.

Two authors (K.Z.K.O. and K.-Y.C.) independently extracted data from included studies using a standardised evidence table. Extracted data included the first author’s name, year of publication, study population, country of origin, study design, number of patients, and incidence of laryngospasm.

Included studies were appraised by two authors (K.Z.K.O. and K.-Y.C.) using the JBI Critical Appraisal Tool for cohort studies and randomised controlled trials [26]. These studies were scored against each item on the checklist with ‘Yes’, ‘No’, or ‘Not Sure’.

The literature results were narrated qualitatively. Search numbers, characteristics of the included studies, quality appraisals of the studies, and major findings were summarised and tabulated. Due to the number of studies included and the lack of incidence values in certain studies, meta-analysis was not performed to synthesise the findings.

## 3. Results

### 3.1. Study Selection

From the literature search, 26 potentially eligible records were identified. Three duplicates were removed, leaving 23 studies screened for titles and abstracts. Subsequently, seven studies, which were not RCTs and cohort studies, were excluded, followed by two studies that were not published in English. The full texts of fourteen studies were reviewed, and nine studies were excluded because they did not meet our inclusion criteria (three studies did not involve adenotonsillectomy, three studies did not involve the paediatric population, two studies did not collect incidences of laryngospasm, and one study did not compare the LMA and ETT). Finally, five studies were included in our final analysis. The process for the literature search and study selection is presented in Figure 1.

### 3.2. Study Characteristics

Most of the studies reviewed were conducted in the United States. A total of five studies comprising three RCTs and two retrospective cohorts were taken into consideration. The largest study sample obtained was a retrospective analysis conducted in Germany with 1534 subjects [24] (Table 1). All studies included used purely inhalational anaesthesia.

Four studies reported incidences of laryngospasm postoperatively, whereas only one article by Gehrke et al. [24] documented intraoperative laryngospasm incidences.

### 3.3. Quality Assessment

Quality assessment of the studies included revealed a generally low risk of bias. For RCTs (Table 2), allocation concealment and blinding of patients, administrators, and assessors were impossible because the LMA and ETT are two distinct techniques that cannot be blinded. Webster et al. [11] indicated that patients were randomised, but the method of randomisation was not disclosed. Otherwise, all RCTs shared a low risk of selection and measurement bias. Similarly, the risk of bias was low for the cohort studies (Table 3). Although data on confounders were collected, the chi-square test used to analyse the incidence of laryngospasm did not permit adjustment of confounders as it would in multivariate tests.
jcm-14-03369-t002_Table 2Table 2Critical appraisal for RCTs included.Study/Item12345678910111213Webster et al. 1993 [11]NSNYNNNYYYYYYYPeng et al. 2011 [8]YNYNNNYYYYYYYSierpina et al. 2012 [13]YNYNNNYYYYYYYNotes: N, No; NS, Not sure; Y, Yes.

Items:Was true randomisation used for assignment of participants to treatment groups?Was allocation to treatment groups concealed?Were treatment groups similar at the baseline?Were participants blind to treatment assignment?Were those delivering treatment blind to treatment assignment?Were outcomes assessors blind to treatment assignment?Were treatment groups treated identically other than the intervention of interest?Was follow-up complete, and if not, were differences between groups in terms of their follow-up adequately described and analysed?Were participants analysed in the groups to which they were randomised?Were outcomes measured in the same way for treatment groups?Were outcomes measured in a reliable way?Was appropriate statistical analysis used?Was the trial design appropriate, and any deviations from the standard RCT design (individual randomisation, parallel groups) accounted for in the conduct and analysis of the trial?
jcm-14-03369-t003_Table 3Table 3Critical appraisal for cohort studies included.
1234567891011Webb et al. 2021 [17]YYYYNSYYYYYYGehrke et al. 2019 [24]YYYYNSYYYYYYNotes: NS, Not sure; Y, Yes.

Items:Were the two groups similar and recruited from the same population?Were the exposures measured similarly to assign people to both exposed and unexposed groups?Was the exposure measured in a valid and reliable way?Were confounding factors identified?Were strategies to deal with confounding factors stated?Were the groups/participants free of the outcome at the start of the study (or at the moment of exposure)?Were the outcomes measured in a valid and reliable way?Was the follow-up time reported and sufficient to be long enough for outcomes to occur?Was follow-up complete, and if not, were the reasons to loss to follow-up described and explored?Were strategies to address incomplete follow-up utilised?Was appropriate statistical analysis used?

### 3.4. Study Results

#### 3.4.1. Incidence of Laryngospasm

Webb et al. [17] revealed that three patients required conversion to ETT due to the pooling of blood, which triggered laryngospasm. Peng et al. [8] passed a flexible fibreoptic scope upon conclusion of surgery in all their LMA group patients. They noted that only one case had blood present at the laryngeal inlet, but the patient did not experience any post-extubation laryngospasm or desaturation. No differences in incidences of laryngospasm were observed between the two groups.

Webster et al. [11] found that laryngospasm occurred in six patients in the ETT group, as opposed to three in the LMA group, but attributed it to the presence of the oropharyngeal airway, which triggered the laryngeal response upon insertion after the removal of the LMA.

A retrospective study with a larger population (*n* = 1534) by Gehrke et al. [24] showed a similar ratio of postoperative laryngospasm between both groups. Two patients in the study by Sierpina et al. [13] experienced brief episodes of laryngospasm in the LMA group, which were easily managed using face mask ventilation for <1 min without any sequelae. There were episodes of laryngospasm in the ETT group. However, this finding was not statistically significant (Table 4).

#### 3.4.2. Rates of Conversion from LMA to ETT

Intraoperative conversion to ETT was necessary in 75 cases, representing 10.96% of all patients treated with an LMA in the study of Gehrke et al. [24]. In the study of Webb et al. [17], 1.2% of cases required conversion to ETT overall, consisting of 11 of 688 (1.6%) adenotonsillectomies and 1 of 325 (0.3%) adenoidectomies. Reasons for conversion in Webb et al.’s study included laryngospasm (three cases) and poor fit/visualisation (nine cases). In the prospective trial by Peng et al. [8], 10 of 60 children initially assigned to the LMA (16.7%) were converted to ETT intraoperatively due to kinking of the tube or poor visualisation. An additional two children initially assigned to the LMA group were intubated with an ETT from the start due to bronchospasm during mask induction. One study reported no instances of conversion from the LMA to ETT in their 65 LMA cases [13].

## 4. Discussion

This study revealed that incidences of laryngospasm between the LMA and ETT in paediatric adenotonsillectomies were similar. In the two RCTs included, LMAs did indeed reduce the incidence of laryngospasm [8,11]. In contrast, two retrospective studies revealed that ETT was more advantageous in reducing incidences of laryngospasm, but the difference was not statistically significant [17,24].

In the study by Gehrke et al. [24], the authors did not support the use of LMA as an airway device for paediatric adenotonsillectomies. Their opinion was mainly based on a significant increase in surgical duration, an unfavourable perioperative safety profile, and the need to change the LMA to ETT in 10% of their patients. Many studies have attributed the prolonged surgery time to difficulties in visualising the surgical field, which is secondary to the size and shape of the LMA [24]. Kinking and leaking during the introduction of the mouth gag, leading to ventilation difficulties and subsequently needing a change to ETT, has been reported as well [7,8,27,28]. The induction time was shorter for the LMA, but was not statistically significant, which could be explained by the fact that even though LMA insertion does not require visualisation of the larynx, endotracheal intubation in children is generally very fast as well. Studies that reported a significant time reduction needed for emergence from anaesthesia were conducted in institutions where patients were transferred to recovery with the LMA still in place, while extubation of endotracheal tubes was performed in the operating theatre [7,11].

A systematic review and meta-analysis of 12 studies, including 4176 patients, suggested that the LMA is a safe alternative to ETT in adenotonsillectomies. Although the conversion rate from the LMA to ETT was 8.27% in the paediatric subgroup, it was not due to adverse perioperative respiratory events, but due to surgical access. Therefore, the authors recommend that the careful selection and judgement of surgeons and anaesthesiologists is necessary [29].

A meta-analysis by Luce et al., which included 19 RCTs, noted that the incidence of laryngospasm was affected by the depth of anaesthesia upon removal of the LMA [30]. It was noted that a deeper plane of anaesthesia upon removal is more favourable in reducing the stimulating effect of LMAs on the upper respiratory tract, potentially triggering the laryngospasm reflex, which is particularly strong in children [31,32].

Another meta-analysis conducted by Patki [33] did not find that LMAs offered any advantage in reducing the incidence of laryngospasm. They concluded that respiratory complications, i.e., laryngospasm, could be multifactorial. The lack of complete information regarding the possibility of improper endotracheal tube size and possible complications during insertion of LMAs could be confounding factors that may have modified the actual interpretation of results in that meta-analysis [34].

The findings of this systematic review cannot be applied to patients with abnormal airways, as all subjects recruited were of ASA category I or II. Furthermore, airway manipulations were performed by paediatric anaesthesiologists and surgeries were performed by paediatric otolaryngologists, which does not reflect the actual clinical scenario in hospitals, as adenotonsillectomy is usually performed by junior surgeons and anaesthesiologists. A larger cohort and a broader inclusion criterion would probably accurately replicate real-world data for incidences of laryngospasm in paediatric adenotonsillectomy.

In addition, this review only included studies that were published in English, which may have limited the actual data representation. Similarly, single-centred studies may tend to underrepresent the overall population. The small number of studies included, and the shortage of randomised controlled trials were also limiting factors of this systematic review. Due to the small number of studies, we also included the study of Gehrke et al., which included patients who underwent adrenalectomy, tonsillectomy, or both. This inclusion introduced some heterogeneity in this review, despite the majority of subjects having undergone adenotonsillectomy (64.0%). Possible confounding biases towards ETT or LMAs, such as familiarity or preference towards one or the other by both surgeons and anaesthetists, and the child’s behaviour during induction could have contributed to excessive secretions from crying, causing postoperative laryngospasm. Studies involving the depth of anaesthesia during the removal of airway devices and their correlation with the incidence of laryngospasm should be studied as well.

## 5. Conclusions

The LMA or ETT does not affect the incidence of laryngospasm in paediatric adenotonsillectomy. A multi-centred RCT comprising paediatric patients with different comorbidities and different levels of surgical and anaesthetic experience will fill in the research gap and further the investigation of whether one device is superior to the other.

## Figures and Tables

**Figure 1 jcm-14-03369-f001:**
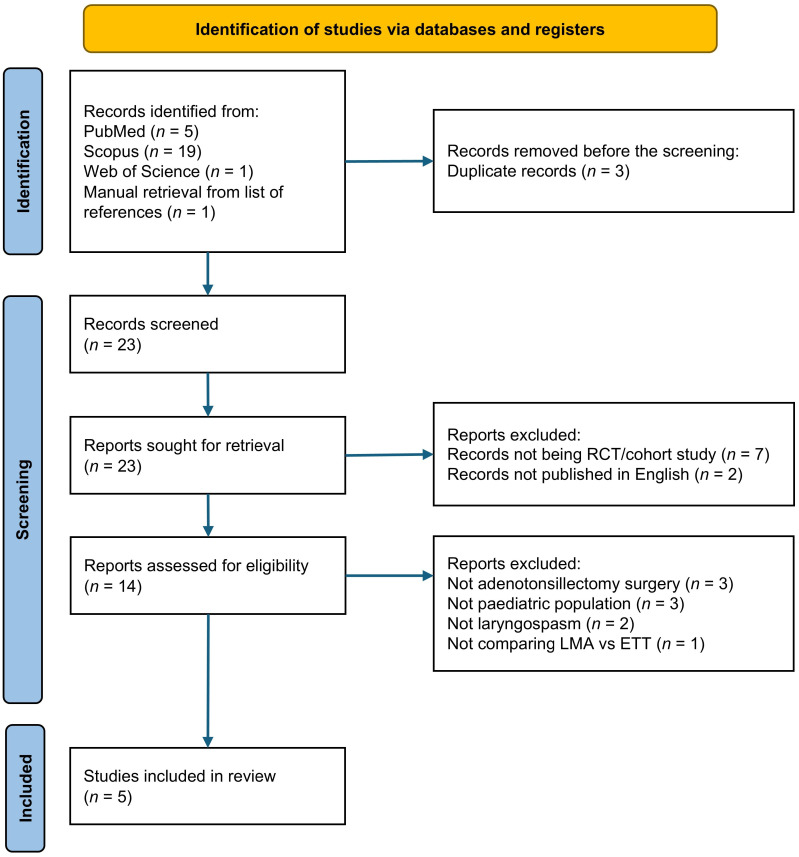
PRISMA flow chart.

**Table 1 jcm-14-03369-t001:** Characteristics of studies included.

Author	Country	Year of Publication	Sample Size (n)	Study Design
Peng et al. [8]	USA	2011	131	RCT
Sierpina et al. [13]	USA	2012	117	RCT
Webster et al. [11]	Canada	1993	109	RCT
Webb et al. [17]	USA	2021	1042	Retrospective
Gehrke et al. [24]	Germany	2019	1585 *	Retrospective

* Only 1015/1585 patients had both an adrenalectomy and tonsillectomy; the rest underwent only one of these procedures.

**Table 4 jcm-14-03369-t004:** Incidences of laryngospasm in different groups in the studies included.

Author	Study	Age (Expressed as Mean (SD), Except Median (Range) in Gehrke et al.)		LMA (Flexible LMA)		ETT		*p*-Value
		LMA	ETT	Laryngospasm Incidence	Total	Laryngospasm Incidence	Total	
Peng et al. [8]	RCT	5.69 (2.30)	5.55 (2.50)	6	48	8	83	0.77
Sierpina et al. [13]	RCT	5.02 (2.31)	5.88 (2.33)	ND	65	ND	50	0.504 ^#^
Webster et al. [11]	RCT	4.99 (1.69)	4.38 (1.85)	3	55	6	54	0.49
Webb et al. [17]	Retrospective Cohort	4.99	4.85	3	100	0	100	ND
Gehrke et al. [24]	Retrospective Analysis	4.374 (0.4–16.9)	4.596 (0.6–17.0)	8 (0.88%)/ 4 (0.58%)	683	3 (0.42%)/ 2 (0.24%)	849	0.06 *

ND, not disclosed. ^#^ Bonferroni correction level with chi-square was performed in this study. * Intraoperative and postoperative laryngospasm were studied, and an average *p*-value was obtained.

## Data Availability

Not applicable.

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
