# Peer review of "Incidences of Laryngospasm Using a Laryngeal Mask Airway or Endotracheal Tube in Paediatric Adenotonsillectomy: A Systematic Review"

_jcm, 2025, doi:10.3390/jcm14103369_

Round 1
Reviewer 1 Report
Comments and Suggestions for Authors
In the manuscript “A Systematic Review on the Incidence of Laryngospasm between the Use of Laryngeal Mask Airway and Endotracheal Tube in Paediatric Adenotonsillectomy”” the authors review pertinent literature evaluating the incidence of laryngospasm in patients receiving LMAs versus endotracheal tubes (ETT) for T+A.
I congratulate the authors for presenting a review of the current evidence of LMA vs ETT use for T+As as this is becoming an increasingly popular subject. This works concludes that LMAs do not reduce the incidence of laryngospasm as compared to ETT in pediatric T+A. This study lays important groundwork for continued randomized control trials to evaluate this subject as LMAs for T+A becomes progressively utilized.
Specific Comments:
-Page 2, line 49-59: I think it is important to mention the conversion rate from LMA to ETT in these procedures due to poor surgical visualization.
-Page 6, table 4: It would be beneficial to include the type of LMA used in each study as this may impact laryngospasm rates. Also, please include information on the type of anesthesia used; inhalational vs propofol TIVA.
-Page 7, line 260-267: Another limitation is the small number of studies included and the shortage of randomized control trials included
Author Response
Thank you for reviewing our manuscript. Please kindly refer to the attachment for our response.

Reviewer 2 Report
Comments and Suggestions for Authors
Despite its aim to address an important clinical question, this systematic review is methodologically weak and conceptually inconsistent. The inclusion of Gehrke et al.'s study, which analyzed adenoidectomies only, introduces significant clinical heterogeneity and undermines the focus of the review. The failure to stratify by procedure type, airway anatomy, or patient comorbidities, combined with a shallow narrative synthesis and the absence of a meta-analysis, makes the conclusions tenuous. Overall, the review adds limited new insight and does not meet the rigor expected of a systematic review in a high-impact clinical journal.
Author Response

(The authors gave the same response as above.)

Reviewer 3 Report
Comments and Suggestions for Authors
The authors reviewed a number of papers to determine whether the use of LMA or ETT for adenotonsillectomy was superior in terms of avoiding laryngospasm and found essentially no difference.
The authors compared the incidence of Laryngospasm Be-tween the Use of Laryngeal Mask Airway and Endotracheal Tube in Pediatric Adenotonsillectomy. I have not perceived this as a gap in the field, but I am an ENT surgeon and not an anesthesiologist. The results imply that there may be a decreased incidence of laryngospasm by using an LMA but these are lessened by the fact that bleeding may result in a need to switch to an ETT tube in certain cases, negating that result. The methodology seems sufficient to this reviewer. The conclusions seem consistent with the evidence and arguments presented. The comparison is valid. The references are appropriate.
Author Response

(The authors gave the same response as above.)

Round 2
Reviewer 2 Report
Comments and Suggestions for Authors
The authors have introduced substantial improvements that clearly enhance the quality of the manuscript. The revised title is more accurate and better aligned with the content of the study. The introduction and abstract are now more coherent, and the methodology section has been clarified, adhering more closely to PRISMA guidelines and including protocol registration. The inclusion and exclusion criteria are better defined, and the discussion now offers a more thoughtful analysis of the study’s limitations.